# Fungal Endophthalmitis: A Comprehensive Review

**DOI:** 10.3390/jof7110996

**Published:** 2021-11-22

**Authors:** Abid A. Haseeb, Abdelrahman M. Elhusseiny, Mohammad Z. Siddiqui, Kinza T. Ahmad, Ahmed B. Sallam

**Affiliations:** 1Department of Ophthalmology and Visual Sciences, University of Illinois at Chicago, Chicago, IL 60612, USA; haseeb2@uic.edu; 2Department of Ophthalmology, Harvey and Bernice Jones Eye Institute, University of Arkansas for Medical Sciences, Little Rock, AR 72205, USA; AMElhusseiny@uams.edu (A.M.E.); MSIDDIQUI@uams.edu (M.Z.S.); ahmadkinza@gmail.com (K.T.A.)

**Keywords:** fungal endophthalmitis, exogenous endophthalmitis, endogenous endophthalmitis, *Candida*, *Aspergillus*, antifungals, pars plana vitrectomy

## Abstract

Endophthalmitis is a serious ophthalmologic condition involving purulent inflammation of the intraocular spaces. The underlying etiology of infectious endophthalmitis is typically bacterial or fungal. The mechanism of entry into the eye is either exogenous, involving seeding of an infectious source from outside the eye (e.g., trauma or surgical complications), or endogenous, involving transit of an infectious source to the eye via the bloodstream. The most common organism for fungal endophthalmitis is *Candida albicans.* The most common clinical manifestation of fungal endophthalmitis is vision loss, but other signs of inflammation and infection are frequently present. Fungal endophthalmitis is a clinical diagnosis, which can be supported by vitreous, aqueous, or blood cultures. Treatment involves systemic and intravitreal antifungal medications as well as possible pars plana vitrectomy. In this review, we examine these essential elements of understanding fungal endophthalmitis as a clinically relevant entity, which threatens patients’ vision.

## 1. Introduction

Endophthalmitis refers to inflammation of the internal ocular structures with involvement of the vitreous and aqueous humors [1,2,3]. While some definitions in the literature characterize it as any inflammation of the intraocular spaces [3], in clinical practice it typically refers to inflammation secondary to an infectious bacterial or fungal cause [1,2,3,4]. If infection spreads from the globe to the neighboring soft tissues within the orbit, it is then classified as panophthalmitis [1]. Endophthalmitis is subdivided into exogenous and endogenous forms. Exogenous endophthalmitis refers to a condition in which the infectious source is external to the eye. Possible causes of exogenous endophthalmitis include intraocular surgery, penetrating trauma, and contaminated intraocular foreign bodies [2,3,5]. Conversely, endogenous endophthalmitis is less common and occurs secondary to hematogenous spread of an infectious source into the eye [6,7,8]. Also, in contrast to exogenous endophthalmitis, endogenous endophthalmitis is more closely associated with distinct medical risk factors, namely those which increase the likelihood of systemic infection such as immunocompromising conditions and diabetes mellitus [9,10,11,12,13,14]. Endophthalmitis is a serious vision-threatening condition, and knowledge of its diagnosis, clinical presentation, and management is essential for both ophthalmologists and physicians dealing with infectious disease.

In general, the prevalence of fungal endophthalmitis, both endogenous and exogeneous, is lower than bacterial endophthalmitis. This has led to absence of level 1 evidence to guide treatment. The purpose of the current review is to provide a comprehensive updated literature about the diagnosis and management of fungal endophthalmitis.

## 2. Etiologies and Pathogenesis

### 2.1. Exogenous Endophthalmitis: Overview

Exogenous endophthalmitis was first described in 1933 by Rychener, who divided it into the categories of contiguous spread from an external ocular infection, penetrating trauma, and intraocular surgery [15]. Pflugfelder’s later report on exogenous fungal endophthalmitis identified it in cases of ocular surgery, trauma, keratitis, and infection of a filtration bleb [5]. Overall, exogenous endophthalmitis is the most common type of endophthalmitis, accounting for upwards of 80% of cases [1,2,16]. No high-powered studies have examined the relative incidence of bacterial versus fungal etiologies in exogenous endophthalmitis overall; data are generally limited to subclassifications of exogenous endophthalmitis, such as an overwhelming majority of bacterial sources in acute post-cataract surgery endophthalmitis (approaching 100% in the United States and Europe [1,2,17]) and a fungal majority (approximately 50% *Aspergillus* and *Fusarium* species) in keratitis-related exogenous endophthalmitis [2]. In terms of speciation, in a study examining 47 isolates from patients with exogenous fungal endophthalmitis, Silva et al. identified that 14 (29.8%) were caused by *Candida* species, 10 (21.3%) by *Fusarium* species, and 8 (17.0%) by *Aspergillus* species [18]. Among fungal causes of exogenous endophthalmitis, estimates for incidence vary. However, Wykoff et al. studied 41 cases of culture-positive exogenous fungal endophthalmitis and found that 18 cases (44%) were associated with fungal keratitis, 10 cases (24%) occurred secondary to penetrating ocular trauma, and 13 cases (32%) occurred after intraocular surgery [17]. Similar to Silva et al., they found that filamentous fungi (molds)—primarily *Aspergillus* and *Fusarium* species—accounted for 35 cases (85%), and *Candida* species (yeasts) accounted for the remaining 6 cases (15%). 

Risk factors for the development of exogenous fungal endophthalmitis are not well-studied, as unlike endogenous endophthalmitis, there is no strong association with underlying medical conditions or systemic disease. Risk factors for exogenous endophthalmitis correspond with the type of exogenous cause. For example, in keratitis-associated exogenous fungal endophthalmitis, risk factors include contact lens use, trauma with organic matter, and LASIK [17].

#### 2.1.1. Exogenous Endophthalmitis: Traumatic Endophthalmitis

Post-traumatic endophthalmitis is an uncommon but possible complication of open globe injury secondary to foreign bodies, lens rupture, or trauma with contaminated objects (Figure 1) [19]. Essex et al. studied 250 cases of patients admitted to a single hospital with open globe injuries during a three-year period and found that the frequency of endophthalmitis was 6.8%. This is comparable to the more recent findings of Tan et al. who found 26 endophthalmitis cases amongst 448 open globe injuries (5.8%) [20]. Essex et al. determined that the following were all significantly associated with the development of endophthalmitis in the setting of open globe trauma: dirty wounds, retained foreign bodies, lens capsule breach, delayed primary repair, and a rural address [21].

Regarding the causative pathogens, Long et al. studied 912 cases of post-traumatic endophthalmitis and found that 38.1% cases were culture positive, and 3.2% had mixed infections of Gram-negative bacilli and fungi [22]. Of culture positive organisms, 41.9% were Gram-positive cocci, 29.1% were Gram-negative bacilli, and 16.8% were fungi. These organisms are summarized in Table 1.

#### 2.1.2. Exogenous Endophthalmitis: Other Surgical and Procedural Causes

Acute post-cataract endophthalmitis is an important cause of exogenous endophthalmitis. The majority of cases in the United States and Europe are caused by coagulase-negative staphylococci (70%) and other Gram-positive cocci (25%) [2,23,24]. In more tropical locations such as India, a fungal predominance of pathogenesis is seen (21.5% compared to 10.0% confirmed bacterial positivity) in post-cataract endophthalmitis [25]. Narang et al. found that in 27 cases of post-cataract fungal endophthalmitis at a tertiary care hospital in India with a positive vitreous aspirate, 16 (59.3%) were *A. flavus*, 3 (11.1%) were *A. fumigatus*, and the rest included *A. niger*, *Acremonium kiliense*, *Fonsecaea predosoi, Candida guilliermondii*, *C. albicans*, and *C. tropicalis* [26].

Another rare surgical cause of exogenous endophthalmitis is penetrating keratoplasty (PKP) (Figure 2). The reported incidence overall is lower than 1% and has been on the decline [27,28]. One study which examined infectious causes of endophthalmitis after PKP found that in cases of culture-positivity, 30.7% of cases were fungal, of which the *Candida* species were the most common [28]. A study by Keyhani et al. at the New York Eye and Ear Infirmary found that of 344 donor corneas collected over the course of five years which had positive rim cultures, 28 (8.6%) were positive for fungi, all of which were *Candida* species, the most common of which were *C. albicans* (75%) and *C. glabrata* (18%) [29]. Overall, there was incidence of post-surgical fungal endophthalmitis of 0.16%. Similarly, while exceedingly rare, case reports have indicated the possibility of fungal endophthalmitis following Descemet stripping automated endothelial keratoplasty (DSAEK) surgery via the presence of surface venting incisions. Wang et al. described this finding in an 85-year-old patient who was found to have bleb purulence and corneal infiltrates after DSAEK [30], and Chew et al. reported this finding in a 72-year-old woman who developed stromal and intraocular inflammation after DSAEK [31]. Both cases were due to presumed *Candida* infection.

Due to the increased use of intravitreal injections as a therapeutic modality in ophthalmology, endophthalmitis associated with intravitreal injections has emerged as an important consideration. From an epidemiological standpoint, the risks of fungal endophthalmitis secondary to intravitreal injection are low. Multiple large studies performed studying this complication have revealed an endophthalmitis rate of 0.02–0.05% in general; in these studies, no cases of fungal endophthalmitis were reported [16,33,34]. However, Sheyman et al. reported on a case series of eight eyes of eight patients all developing fungal endophthalmitis secondary to injections contaminated with *Exserohilum* and *Bipolaris hawaiiensis*, underscoring the importance of using rigorously-protocolized injection samples [35]. These cases were concluded to be secondary to contamination at the preparing pharmacy.

Although exceedingly rare, fungal endophthalmitis has been reported following trabeculectomy for glaucoma as well [36,37].

#### 2.1.3. Exogenous Endophthalmitis: Keratitis-Associated Endophthalmitis

Fungal keratitis is a serious cause of corneal blindness. In developed countries, fewer than 10% of infectious keratitis cases are fungal. Green et al. examined 253 cases of microbial keratitis over a five-year period in Australia and found fungal positivity in 3% of cases, with *Fusarium* being the most common [38]. Similarly, Ritterband et al. studied 5083 cases of infectious keratitis over a 16-year period and found fungal positivity in 61 (1.2%) cases; of these, *C.* albicans (47.5%) was the most common [39]. Human immunodeficiency virus (HIV) seropositivity, chronic ocular surface disease, and trauma were identified as risk factors.

Fungal keratitis is more common in developing countries and tropical climates; for example, Sirikul et al. found fungal etiologies in 37.7% of infectious keratitis in Thailand [40], and in the Delhi Infectious Keratitis Study this figure was 36.4% [41]. A ten-year review of 1352 cases in southern India found it to be significantly more common in males, younger patients, and during the monsoon and winter seasons [42]. Ocular trauma predisposed to infection in 54.4% of eyes, consistent with the figure of 42.1% reported by Joseph et al. [43]. The most common fungal pathogens were *Fusarium* (37.2%) and *Aspergillus* (30.7%) species. Fungal keratitis is the most common infectious corneal disease in China and is a frequent indication for corneal transplantation [44,45,46,47]. Xie et al. found fungal keratitis to account for 61.9% of cases of severe infectious keratitis over a six-year period in China [45]. Risk factors were identified to be corneal trauma (51.4% of patients) and since it was more common due to injury from plants (25.7%), it was more common in harvest seasons. The most common organisms were *Fusarium* species (73.3%) and *Aspergillus* species (12.1%).

With respect to the risk factors of keratitis-related endophthalmitis, Wan et al. investigated 392 cases of fungal keratitis and found endophthalmitis in 37 (9.4%) patients [44]. Multivariate analysis revealed risk factors for the development of fungal endophthalmitis to be topical steroid use (odds ratio [OR] = 6.35), previous corneal laceration suturing (OR = 5.05), large corneal ulcer size (OR = 4.43), hypopyon (OR = 11.05), and aphakia (OR = 15.45). Corneal perforation is not a sensitive finding for endophthalmitis: In Wan et al.’s study, it was only present in 4 of 37 (10.8%) patients with fungal endophthalmitis in the setting of fungal keratitis. This number was similar (35 of 355, 9.9%) in cases of fungal keratitis without endophthalmitis. Causative pathogens included the *Fusarium* species (15 eyes, 40.5%), *Aspergillus* species (6 eyes, 16.2%), *Alternaria* species (4 eyes, 10.8%), other fungi (9 eyes, 24.3%), and one unidentifiable fungal species (1 eye, 2.7%) [44]. 

### 2.2. Endogenous Endophthalmitis

Endogenous endophthalmitis refers to the development of endophthalmitis secondary to hematogenous dissemination of an infectious insult [2,7,44]. Physicians should have high clinical suspicion of endophthalmitis in the setting of a known mechanism for fungal penetration into the bloodstream, irrespective of a history of immunosuppression [7]. Due to an increased amount of blood flow directed to the choroidal space and ciliary body, endogenous endophthalmitis primarily affects these areas of the eye, beginning with the choroid and then progressing to secondary impact on the retina and vitreous spaces [4,48]. Endogenous endophthalmitis accounts for approximately 2–15% of all cases of endophthalmitis, including both bacterial and fungal etiologies [4,12].

The risk factors for endogenous endophthalmitis are numerous. Immunosuppression per se is not sufficient as a risk factor for fungal endophthalmitis. Instead, immunosuppression must also coincide with a source of fungemia. By definition, any condition predisposing to fungemia may predispose to the development of endogenous fungal endophthalmitis [7,49]. As a result, immunosuppressive conditions compounded by prolonged hospital stays, indwelling intravenous catheters, prolonged or broad-spectrum antibiotic use, and granulocytopenia resulting from intensive chemotherapy all predispose an individual to endogenous endophthalmitis [50,51]. Chakrabarti et al. reported that, in their group of 12 patients with endogenous fungal endophthalmitis, 5 patients (41.7%) had uncontrolled diabetes and 2 (16.7%) had a history of intravenous drug use [52].

An important contributing cause of endogenous endophthalmitis is intravenous drug use: Mir et al. reviewed 56,839 cases of endogenous endophthalmitis hospitalizations in the United States and found that 13.7% had a history of drug dependence or use. If these, 9.8% had *Candida* infection and 1.8% had disseminated candidiasis [53]. Furthermore, the incidence of endogenous endophthalmitis associated with drug dependence or use has increased from 0.08 per 100,000 people in 2003 to 0.32 per 100,000 people in 2016 [53]. This is concurrent with an overall increase in drug-dependence in the United States, particularly secondary to opioid use. In the last decade, there has been a two-fold increase in deaths secondary to drug dependence [53].

Other important risk factors for the development of endogenous endophthalmitis include sepsis, malignancy, hepatitis B, hepatitis C, pneumonia, and urinary tract infection. It is of note that, despite the very high incidence of mucosal candidiasis, endophthalmitis due to candidemia in patients with HIV infection or acquired immune deficiency syndrome (AIDS) is very uncommon in the absence of other risk factors [4,53,54]. *Candida* endophthalmitis has been reported in premature infants with systemic candidiasis [55]. Fungal endogenous endophthalmitis is more common in the Western hemisphere compared to the Eastern hemisphere, although no epidemiologic mechanism for this has been described [10].

Though high-powered studies have not been conducted, limited reports suggest that prolonged corticosteroid use and protracted hospital courses during the severe acute respiratory syndrome coronavirus 2 (SARS-CoV-2, or COVID-19) pandemic may yield an increase in endogenous fungal endophthalmitis cases. Shroff et al. reported on a series of five cases of endogenous fungal endophthalmitis (five *Candida*-mediated, one *Aspergillus*-mediated) in five patients who had been hospitalized for COVID-19 pneumonia for an average of 42 days and had received systemic corticosteroid therapy [56]. All eyes underwent pars plana vitrectomy with intravitreal antifungal therapy with good anatomic and visual response. Though speciation information was not reported, Shah et al. similarly reported on a series of four cases of presumed endogenous fungal endophthalmitis in patients with COVID-19 pneumonia [57].

With respect to causative organisms, reports in the literature are varied. Binder et al. studied 27 cases of endogenous endophthalmitis and found 13 patients (48.1%) with fungal infections, 13 (48.1%) with bacterial infections, and one (3.7%) with mixed bacterial and fungal infection [58]. Of the 14 total patients with fungal infections, 10 had *C. albicans* infection (71.4%) and 4 (28.6%) had *A. fumigatus* infection. In their study of 56,839 cases of endogenous endophthalmitis, Mir et al. identified that *Candida* infection was present in 6.7% of cases, disseminated candidiasis in 1.4% of cases, and *Aspergillus* infection in 0.4% of cases. In their study of 57 cases, Lim et al. found that the most common infectious etiology was Gram-negative bacteria (31 cases, 54.4%), while *C. albicans* was the most common fungal etiology, present in 9 cases (15.8%) [12]. Amongst intravenous drug users, fungal pathogenesis has been identified in as high as 59% of cases, with *C. albicans* being the most common etiology [59,60]. Other possible organisms include *C. dulineniesis*, *C. tropicalis*, and *A. niger* [13,59,60]. Endogenous fungal endophthalmitis secondary to *Histoplasma capsulatum* is exceedingly rare but has been reported; Gonzales et al. reported endogenous *H. capsulatum* endophthalmitis with severe subretinal exudation, choroidal granulomas, and intraretinal hemorrhage leading to bilateral exudative retinal detachments in a 30-year-old with AIDS, previous pulmonary tuberculosis, previous cerebral toxoplasmosis, and AIDS dementia [61].

## 3. Clinical Presentation

The clinical presentation of fungal endophthalmitis involves floaters or decreased vision in both endogenous and exogenous forms. In endogenous endophthalmitis, both eyes can be involved, and the onset is usually subacute. With respect to visual acuity at presentation, Chakrabarti et al. reported that nearly all their cases of fungal endophthalmitis (53 postoperative, 48 post-traumatic, and 12 endogenous) presented with visual acuity worse than hand motions [52]. More specifically, for patients with endogenous fungal endophthalmitis, visual acuity at presentation was worse than hand motions in 100% of their 12 patients [52]. Other than floaters and decrease vision, the eye can be painful if there is significant iritis or keratitis. Since endogenous endophthalmitis frequently involves systemic disease and necessarily involves hematogenous spread, systemic symptoms such as fevers may be concurrent in endogenous endophthalmitis [2,53,62].

The clinical presentation of fungal endophthalmitis is variable. Peripheral fungal lesions may be asymptomatic and discovered with patient’s referral for ocular consultation based on positive blood culture or diagnosis of systemic fungal infection. Ocular examination may show eyelid edema, conjunctival injection, anterior chamber inflammation with or without a hypopyon, absent red reflex, and vitreous exudation. Intraretinal hemorrhages, nerve fiber layer infarcts, Roth spots and cotton wool spots are nonspecific findings on fundus examination that may not be directly related to the ocular infection. Fungal chorioretinitis and endophthalmitis due to *Candida* species classically presents with multiple creamy-white or fluffy, well-circumscribed retinal lesions sometimes having a “string of pearls” appearance [7,63]. These lesions are very suggestive of fungal infection and is demonstrated in Figure 3. Importantly, early lesions may be flat in the choroid but progress by protruding into the vitreous cavity and may lead to the appearance of intravitreal “puff ball” abscesses [7]. Imaging findings for a patient with *Candida* chorioretinitis are shown in Figure 4.

The time course for endogenous endophthalmitis may differ from exogenous endophthalmitis: Chakrabarti et al. found mean latent periods of clinical presentation for postoperative and post-traumatic fungal endophthalmitis to be 20 and 7 days, respectively, whereas endogenous fungal endophthalmitis had a mean latent period of 30 days [51]. Furthermore, whereas 30–40% of patients with exogenous fungal endophthalmitis presented within one week, only 3% of patients with endogenous fungal endophthalmitis presented within a week [51]. Chakrabarti et al. also reported that corneal edema was more common in postoperative (18.9%) and post-traumatic (22.9%) cases compared to endogenous cases (8.3%), and hypopyon was considerably less common in endogenous cases (25.0%) compared to postoperative (64.5%) and post-traumatic (62.5%) [51]. Vitreous exudates with no red reflex (grade 5) were comparable in cases of endogenous fungal endophthalmitis (50%) compared to postoperative (58.5%) fungal endophthalmitis but less common than in post-traumatic (85.4%) fungal endophthalmitis.

Severe ocular inflammation with vitreous involvement following intraocular surgery should raise suspicion for post-operative endophthalmitis. In a group of 53 cases of postcataract surgery fungal endophthalmitis, Chakrabarti et al. reported vision worse than hand motions in 50 (94.3%) patients. Anterior chamber exudates and hypopyon were present in 34 (64.5%) patients, fibrinous reaction in 38 (70.6%) patients, and vitreous exudates in 10 (21.3%) patients [52]. Loss of red reflex was present in 12 (22.6%) patients. While ocular inflammation out of proportion to previous surgical trauma or expected course should raise suspicion for post-operative endophthalmitis, examination can vary from minimal anterior chamber inflammation to panophthalmitis, corneal edema, or complete anterior chamber hypopyon [4,64].

With respect to traumatic ocular injuries, suspicion for endophthalmitis should be higher by default in eyes with exaggerated signs of inflammation. Post-traumatic endophthalmitis may develop within hours after an initial insult or may develop weeks later [21]. Dirty wounds and lens capsule rupture should particularly raise concern for the possibility of endophthalmitis [19,20,21]. Of 10 patients with exogenous fungal endophthalmitis following penetrating trauma, Wykoff et al. reported that 7 (70%) had visual acuity of no light perception or pthisis [17]. Post-traumatic fungal endophthalmitis may present with purulent exudate, eyelid edema, chemosis, corneal edema, or hypopyon, vitritis, vitreous opacification, or retinitis [19,21,22]. Slowly progressing inflammation following initial ocular repair may raise the suspicion of fungal endophthalmitis [65].

*Aspergillus*-mediated endophthalmitis is usually more severe as compared to *Candida* endophthalmitis with more extensive areas of retinal necrosis and retinal hemorrhages. Rao et al. examined the clinical and histopathologic features of fungal endophthalmitis from 25 patients who underwent enucleation for endogenous fungal endophthalmitis and found that *Aspergillus* endophthalmitis invaded the walls of the retinal and choroidal vessels, in contrast to the vitreous invasion of *Candida* [66]. These cases also occasionally showed thrombosis in the retinal vasculature with focal hemorrhages, exudates, and occasional retinal necrosis. These features are atypical of *Candida* endophthalmitis. Additionally, *Aspergillus* infections may present with thickened eyelid margins and grayish-white corneal stromal inflammation, both atypical for *Candida* infection [67].

## 4. Diagnosis

### 4.1. General Diagnostic Considerations

Endophthalmitis is fundamentally a clinical diagnosis, which may or may not be supported by testing. Tanaka et al. cultured vitreous fluid from vitrectomy biopsy and found fungal positivity in only 30 (38.0%) of 79 cases of known endogenous fungal endophthalmitis, suggesting that the diagnostic yield of culture may be even lower for fungal endophthalmitis compared to bacterial endophthalmitis [68]. Sallam et al. previously studied 43 eyes from 36 patients with known endogenous *Candida* endophthalmitis and found that only 11 (25.6%) had positive cultures from a needle vitreous tap, suggesting that vitrectomy biopsy provides better results than needle aspiration biopsy [8].

Molecular biology based diagnostic techniques as polymerase chain reaction (PCR) based testing of fungi from aqueous or vitreous samples may have higher diagnostic potential. However, there is limited practicality at present for their use in the hospital settings, and they are liable to processing errors that may affect validity. Anand et al. previously studied 43 intraocular specimens of known cases of fungal endophthalmitis and found culture positivity in 24 (55.8%) cases compared to PCR positivity in 32 (74.4%) [69]. In a comparable study, Sandhu et al. found PCR testing was 69.5% sensitive compared to 13.0% sensitivity using culture methods [70]. 

Blood cultures may be used in the diagnostic workup for fungal endophthalmitis. One study found that of 18 eyes with a diagnosis of endogenous endophthalmitis, blood cultures were positive in 6 (33.3%) eyes [14]. Elsewhere, Okada et al. reported positivity of 75% [71]. In both studies, the authors did not specify the percentage in fungal versus bacterial endophthalmitis.

In terms of imaging modalities, B-scan ultrasonography is a mainstay in the diagnosis of endophthalmitis, particularly when the posterior segment is not visualized, and eyes have self-sealing or previously sutured wounds [4,72]. Classic findings of endophthalmitis on ultrasound include choroidal mass with vitreous strands and membranes with reduced mobility [72]. B-scan ultrasound findings for a patient with *Candida* endophthalmitis are shown in Figure 5. While statistical research has not been performed on the diagnostic yield of ultrasonography, clinical observation suggests that vitreous opacification is a relatively sensitive finding, though ultrasound is not a specific test for endophthalmitis [4]. Also, the presence of a choroidal mass involving the retina and projecting into in the vitreous the context of endophthalmitis is highly suggestive of a fungal pathology.

### 4.2. Differential Diagnosis

Fungal endophthalmitis is a diagnostic challenge, which requires a rigorous history and comprehensive ophthalmic examination. The differential diagnosis for endophthalmitis is extensive. This may include non-infectious causes of inflammation or uveitis, including sarcoidosis, Behçet syndrome, sympathetic ophthalmia, and Vogt-Koyanagi-Harada disease. Infectious etiologies include bacterial endophthalmitis, tuberculosis, syphilis, herpes viruses, and toxoplasmosis, as well as masquerade syndromes such as lymphoma.

A combination of disease course, clinical presentation, and consideration of patients underlying systemic commodities help eliciting the diagnosis of fungal endophthalmitis in most cases. This may be supported by the findings of blood cultures or vitreous tap cultures. Blood tests such as T-spot test for TB, syphilis, and toxoplasma antibodies, as well as angiotensin converting enzymes for sarcoidosis, are helpful to rule out other uveitis entities. In some cases, with severe vitreous haze and no view of the fundus, diagnostic vitrectomy to view the fundus and take a larger sample of vitreous for a wider array of tests can be helpful to make the diagnosis, and rule out other pathology such as toxoplasma, viral retinitis, and lymphoma.

Characteristics distinguishing bacterial endophthalmitis from fungal endophthalmitis are presented in Table 2. An important clue differentiating fungal endophthalmitis from bacterial endophthalmitis lies in the time course of clinical presentation. Bacterial endophthalmitis is usually more rapidly progressive and intraocular inflammation on ophthalmic examination tends to be more diffuse in cases of bacterial endophthalmitis, whereas fungal endophthalmitis progresses slowly and presents with clusters of inflammation in the aqueous or vitreous chambers [2].

The most common incorrect initial diagnosis for endophthalmitis is non-infectious uveitis [2]. Chen et al. found that, of 51 patients they studied, 12 (23.5%) had an initial negative aqueous or vitreous culture, but later had a positive culture from a PPV specimen. In these patients, there was an initial incorrect diagnosis of non-infectious uveitis [73]. Another group, Shen and Xu, reported that 11 of 20 (55.0%) vitreous specimens obtained via needle tap were negative in patients with endogenous endophthalmitis although PPV specimens later confirmed that all 11 were in fact fungi [74]. Schiedler et al. found that final diagnosis of endogenous *Candida* endophthalmitis was significantly associated with initial incorrect diagnosis of uveitis. In their study examining 21 eyes of 21 patients, 4 patients who initially presented with a diagnosis of uveitis were found to have endogenous *Candida* endophthalmitis [75].

**Table 2 jof-07-00996-t002:** Characteristics distinguishing bacterial endophthalmitis from fungal endophthalmitis.

Characteristic	Bacterial Endophthalmitis	Fungal Endophthalmitis
Frequency and Disease Associations	More common overall [2], and account for the overwhelming majority of post-cataract [2,23,76] and bleb-associated endophthalmitis cases [77]	Less common overall, though accounts for majority of endogenous endophthalmitis cases [2,7,8]
Time Course	Rapidly progressive (days) [2]	Indolent (weeks) [2]
Characteristic Lesions	Diffuse intraocular inflammation [1,2], subretinal abscess can also occue	Clumped appearance of intraocular inflammation [7], choroidal mass projecting into the vitreous chamber [4,7], chorioretinal creamy-white or “string of pearls” lesions (*Candida* species) [7,49]

## 5. Treatment

In general, the mainstay of treatment for endophthalmitis is systemic or intravitreal antibiotics, although surgical management may be indicated. 

### 5.1. Medical Management

#### 5.1.1. Medical Management in *Candida* Endophthalmitis

Treatment of *Candida* endophthalmitis has not been documented in high-powered clinical trials, mainly due to the disease being uncommon. Classically, *Candida* infections have been treated with azole-class drugs including fluconazole and voriconazole, and non-azoles including amphotericin B and caspofungin. However, azole resistance, mainly to fluconazole, is a growing problem in clinical isolates of *Candida* species, limiting their effectiveness [78,79]. Khan et al. performed a systematic review of endogenous *Candida* endophthalmitis cases and found that patients with chorioretinitis respond well to antifungal monotherapy, but in cases of vitreous involvement, drug penetration of amphotericin B is limited and pharmacologic management is insufficient [80].

An understanding of the physiochemical properties of these antifungals clarifies drug penetration. A fundamental problem with amphotericin B is its poor intraocular penetration, which can be understood by its physiochemical properties. It has a relatively high molecular weight of 924 g/mol (approximately three times that of fluconazole (306 g/mol) and voriconazole (349 g/mol)), and it is amphipathic and highly protein bound, all of which are undesirable for transit across the blood-ocular barrier [80]. Voriconazole’s lipophilicity helps explain its superior penetration compared to fluconazole [80].

Of 14 *Candida* isolates examined by Silva et al. in cases of fungal endophthalmitis, all 14 demonstrated susceptibility to intravitreal amphotericin B, oral fluconazole, and voriconazole (oral or intravitreal); however, intravenous amphotericin B levels only reached therapeutic levels in 0–2 (0–14.3%) cases [18]. Voriconazole has previously shown to be superior to fluconazole in its potency against *C. glabrata* [81] and *C. krusei* [82].

Rates of endogenous fungal endophthalmitis and chorioretinitis in the setting of fungemia in older studies were as high as 10–45%, though after improvements in antifungal treatment, prophylactic treatment in high-risk patients, and rapid initiation of treatment for positive blood cultures, more recent rates have consistently been reported as <5% [49,83,84]. The Infectious Diseases Society of America (ISDA) established guidelines for the treatment of disseminated candidiasis include a dilated retinal examination within the first week of therapy, removal of all existing central venous catheters and commencing pharmacologic therapy for a minimum of two weeks following negative blood cultures [1]. Siddiqui et al. previously supported this recommendation by showing that 3 of 161 patients (2%) hospitalized with fungemia were found to have asymptomatic chorioretinitis [49]. However, given the low incidence of ocular infection in candidemia, the utility of ophthalmic consultation on all cases of fungemia has been called into question over the years. Most recently, the American Academy of Ophthalmology released new guidelines in July 2021 that do not support the routine ophthalmologic consultation for candidemia, and instead only screening patients showing symptoms or signs of ocular infection despite treatment of candida septicemia [85]. Despite these guidelines, it is plausible, however, to continue to screen non-communicative patients with candidemia.

For azole-susceptible isolates, the pharmacologic recommendation is a loading dose of fluconazole 800 mg (12 mg/kg) then 400–800 mg (6–12 mg/kg) daily or a loading dose of voriconazole 400 mg (6 mg/kg) intravenous twice daily for 2 doses, then 300 mg (4 mg/kg) intravenous or oral twice daily [86].

For azole-resistant strains, the ISDA recommends liposomal amphotericin B 3–5 mg/kg intravenously daily, with or without oral flucytosine 25 mg/kg four times daily [86]. Liposomal amphotericin B may be less nephrotoxic compared to other formulations and has also been shown to accumulate significantly higher vitreous concentrations compared to amphotericin B deoxycholate (0.47 ± 0.21 vs. 0.16 ± 0.04 µg/mL) [80,87].

Clinically, raised lesions indicate compromise of the blood-retinal barrier, necessitating the delivery of medications directly into the vitreous cavity with intravitreal injections [7]. In these cases, systemic regimen should be combined with intravitreal injection of amphotericin 5–10 µg/0.1 mL or voriconazole 100 µg/0.1 mL [86]. In all instances, duration of systemic treatment should be at least 4–6 weeks depending on anatomic resolution of lesions as determined by an ophthalmologist [86]. Intravitreal injections can be repeated in recalcitrant cases after 72 h.

With respect to efficacy, limited data exist comprehensively examining treatment modalities. Edwards et al. reported a retrospective study examining the efficacy of treatment for *Candida* endophthalmitis and found that 11 of 12 patients (91.7%) were cured with intravenous amphotericin B [88]. An earlier 1995 study by Akler et al. examining cases of *Candida* endophthalmitis found that endophthalmitis was cured in 15 of 16 eyes (93.8%) using oral fluconazole therapy [89]. However, Khan et al. pointed out in their review of pharmacologic agents for the treatment of *Candida* endophthalmitis that only five of these patients had vitritis [80]. In the only randomized, prospective trial we found on our review comparing fluconazole and amphotericin B, Rex et al. found that fluconazole monotherapy 400 mg/day cured 15 of 16 patients (93.8%) with *Candida* endophthalmitis and amphotericin B 0.5–0.6 mg/kg cured 13 of 13 (100%) of patients [90]. In both groups, treatment continued for 14 days after the last negative blood culture. Elsewhere, Filler et al. compared fluconazole and amphotericin B in rabbit models and found that although fluconazole was more effective than saline, amphotericin B reduced fungal colony counts in the vitreous and choroid more than fluconazole, and its fungistatic effects were persistent after 24 days of therapy, unlike fluconazole [91]. That study also found worsening eye lesions in the fluconazole arm. Breit et al. reported on a patient with endogenous *Candida* endophthalmitis who failed treatment with intravenous voriconazole and caspofungin and subsequently was successfully treated with intravitreal amphotericin B [92]. Overall, limited data in the literature suggest that both classes of drugs recommended by the ISDA are efficacious when used appropriately.

Systemic and intravitreal steroids are not routinely used for the management of fungal endophthalmitis, in large part due to the limited data surrounding them and concern about suppressing host immune mechanisms. Majji et al. previously studied 20 cases of culture-proven exogenous fungal endophthalmitis who underwent PPV with intravitreal amphotericin B and oral ketoconazole and found that patients who also received adjunctive intravitreal dexamethasone did not show a statistically significant difference in anatomical and visual outcomes [93]. Coats et al. did examine a model of induced *C. albicans* endophthalmitis in 20 rabbit eyes and found that adjunctive dexamethasone with intravitreal amphotericin B did show statistically-significantly clearer vitreous humor compared to intravitreal amphotericin B alone [94]. No such data have been reproduced in the clinical setting, however.

#### 5.1.2. Medical Management in *Aspergillus* Endophthalmitis

With respect to *Aspergillus* endophthalmitis, ISDA guidelines recommend systemic oral or intravenous voriconazole, plus intravitreal voriconazole or intravitreal amphotericin B deoxycholate [95].

Previously, Silva et al. examined susceptibilities to amphotericin B in 8 isolates and found that all isolates were resistant to intravenous amphotericin B and 7 of 8 (87.5%) were resistant to intravitreal amphotericin B [18]. Conversely, all isolates were susceptible to oral and intravitreal voriconazole. These findings corroborate the recommendations of the ISDA.

In terms of efficacy, Dave et al. studied a series of 91 eyes with *Aspergillus* endophthalmitis (81 [89.0%] exogenous, 10 [11.0%] endogenous) and found that use of intravitreal voriconazole was associated with better visual outcomes (OR = 3.63, CI = 1.2–10.94, *p* < 0.05) but not better anatomical outcomes (OR = 1.12, CI = 0.44–2.83, *p* = *0*.79) when compared to management using systemic antibiotics and no intravitreal injections [96]. Danielescu et al. reported a case of successful inflammatory resolution of *Candida ciferri* post-operative endophthalmitis using intravitreal capsofungin in the setting of *C. ciferri* resistance to fluconazole, voriconazole, and amphotericin B [97].

Limited reports exist on treatment of other etiologies of fungal endophthalmitis. Reports on *Fusarium* species suggest that systemic voriconazole with or without amphotericin B, plus intravitreal injections of voriconazole [98,99,100].

A summary of medical management for fungal endophthalmitis is presented as an algorithm in Figure 6.

### 5.2. Surgical Management

Pars plana vitrectomy is the surgical removal of vitreous humor. In fungal endophthalmitis, there are three levels of indications for the surgery-diagnostic vitrectomy, treatment of infection, and management of surgical complications that result from infection.

The role of PPV to help elicit the diagnosis has been discussed under the diagnosis section in this article. This includes providing an opportunity to clear the vitreous opacity and visualize the chorioretinal pathology, as well as obtaining a large vitreous sample that has a higher yield than a needle vitreous tap [8,49,68].

There are no consensus guidelines on the role of PPV in the treatment of fungal endophthalmitis. It is difficult to ascertain the efficacy of PPV since patients frequently receive PPV contemporaneously with pharmacologic therapy. Removal of the vitreous should at least theoretically decrease the infection load and increase the access of antifungal medications to the retina. However, there is no evidence to confirm that PPV use confers better vision that medical treatment alone. Behera et al. compared visual outcomes in 66 patients with fungal endophthalmitis (56 of 66 [84.8%] exogenous and 10 of 66 [15.2%] endogenous) with immediate PPV versus diagnostic (delayed by a mean of 18.8 ± 10.57 days from presentation) PPV with intravitreal antibiotics. They found that although there was a statistically significant increase in vision in the immediate PPV group, there was no difference in the proportions of eyes reaching a postoperative acuity of ≥20/200 between the two groups [101]. Another retrospective study of 44 eyes with endogenous *Candida* endophthalmitis by Sallam et al., demonstrated that early vitrectomy, within one week of presentation, did not significantly reduce risk of profound visual loss (post-operative Snellen acuity of ≤20/200), yet it decreased the risk of retinal detachment by five-fold [8]. There are other studies that suggest that PPV may improve the anatomical outcome. Celiker et al. reported on seven eyes, which underwent PPV for endogenous fungal ophthalmitis and reported a flat retina in all post-surgical patients, and that no patients developed phthisis bulbi [102]. Similarly, Ghoraba et al. reported on 10 eyes, which underwent PPV for endogenous fungal endophthalmitis and reported a flat retina in all post-surgical patients, with no development of phthisis bulbi [103].

Pars plana vitrectomy also has a role in the treatment of retinal complications of infectious endophthalmitis. These include vitreous opacification, epiretinal membrane formation, and retinal detachment. One study examining 42 eyes with delayed (mean follow-up 48 weeks) consequences of infectious endophthalmitis included 22 (52.4%) with vitreous opacities, 9 (21.4%) with epiretinal membrane, and 11 (26.2%) with retinal detachment [104]. That study found a significant visual acuity improvement from an average of 20/1482 to 20/447, suggesting that PPV may be an important treatment modality both in the acute and chronic setting of fungal endophthalmitis.

### 5.3. Adverse Outcomes in Treatment

Possible complications of treatment modalities for fungal endophthalmitis are numerous. Amphotericin B has a well-studied side effect profile of nausea, vomiting, rigors, fever, hypertension or hypotension, hypoxia, and most notably nephrotoxicity [105,106]. As mentioned previously, data suggest that liposomal amphotericin B formulations may have better nephrotoxicity profiles when compared to other formulations [107,108]. On the other hand, voriconazole and other azole derivatives have a more favorable side effect profile [109,110]. Voriconazole is however associated with a more hepatoxicity risk than fluconazole, warranting careful use in patients with pre-existing liver disease [111,112,113]. Likewise, voriconazole requires more aggressive therapeutic drug monitoring than fluconazole due to the fact that it exhibits classical Michaelis-Menten (non-linear) pharmacokinetics whereas fluconazole exhibits linear pharmacokinetics [114]. Finally, as an inhibitor of CYP3A4, voriconazole has more drug interactions than fluconazole [114].

Complications of PPV in the treatment of fungal infection are well-studied and include secondary cataract, macular edema, post-operative hypotony, pre-retinal membrane, retinal tears, and rhegmatogenous detachment [64,115].

## 6. Visual Prognosis and Complications

### 6.1. Visual Prognosis

The visual prognosis in fungal endophthalmitis is, in general, poor, with exogenous causes portending a worse prognosis than endogenous causes. At a three-month follow-up, Chakrabarti et al. reported that visual outcomes better than 20/400 were more common (66.7%) in endogenous fungal endophthalmitis compared to postoperative (47.2%) and post-traumatic (33.3%) cases [52]. In a study examining 44 eyes of 36 patients with endogenous *Candida* endophthalmitis, Sallam et al. reported sustained vision loss in 23 of 44 (52.3%) eyes at final follow-up, including 16 of 44 (36.4%) with vision worse than 20/200 [8]. Poor initial visual acuity and central lesions were strongly predictive of sustained vision loss. Weishaar et al. examined 12 eyes in 10 patients with endogenous *Aspergillus* endophthalmitis and found sustained vision loss in 8 of 12 (66.7%) of eyes, including 7 of 12 (58.3%) with vision worse than 20/200 [116]. They found that macular involvement was strongly associated with poor visual outcomes. Though sample sizes were low, Schiedler et al. reported that three patients with *Aspergillus* endophthalmitis had a statistically significant worse visual outcomes (all three having visual acuities of light perception or worse) compared to patients with *Candida* endophthalmitis (three of five, or 60%, achieving visual acuity of 20/400 or better) [75]. These latter results corroborate the separate studies of Sallam et al. and Weishaar et al., suggesting that visual outcomes are worse in *Aspergillus*-mediated endogenous endophthalmitis.

### 6.2. Complications

Secondary complications to fungal endophthalmitis are of clinical importance. Wykoff et al. reported that 10 of their 41 patients (24.4%) with exogenous fungal endophthalmitis underwent enucleation as a later procedure, and 7 of 10 (70.0%) open-globe cases underwent enucleation [17]. Additionally, Chen et al. identified a retinal detachment rate of 26% following cases of endogenous fungal endophthalmitis [73]. Similarly, William et al. identified a retinal detachment in 8 of 19 eyes (42.1%) during or following PPV for endogenous fungal endophthalmitis [115]. The authors mention that detachment was present in 2 of 19 (10.5%) eyes at the time of PPV, 2 of 19 (10.5%) developed detachment in the five days following PPV, and in 4 of 19 (21.1%) it developed after an average of five weeks post-PPV. Although there was no association between duration of fungal endophthalmitis symptoms and development of detachment (OR = 0.77; 95% CI: 0.32–1.72), poor baseline visual acuity was associated with increased risk of detachment (OR = 5.86; 95% CI: 1.13–82.02). Of note, Sallam et al. reported that early PPV may be protective against development of retinal detachment in endogenous *Candida* endophthalmitis [8]. Elsewhere, Naoi and Sawada reported on fibrovascular membrane and thick preretinal membrane formation in three patients following PPV for endogenous fungal endophthalmitis [117].

## 7. Discussion

Endophthalmitis is a severe complication of several possible ophthalmic insults. In this review, we sought to explicate the particulars of both endogenous and exogenous forms of fungal endophthalmitis including their manifestations, diagnosis, treatments, and outcomes. It is vital that physicians and ophthalmologists familiarize themselves with this infection entity.

In general, only a limited number of large studies have examined fungal endophthalmitis. From a clinical standpoint, the most important step is establishing the diagnosis of fungal endophthalmitis and starting early antifungal treatment. Collaboration with internists or infectious disease specialists is important in all cases of fungal endophthalmitis to help facilitate the diagnostic workup for systemic fungal infections and to monitor systemic therapy.

## Figures and Tables

**Figure 1 jof-07-00996-f001:**
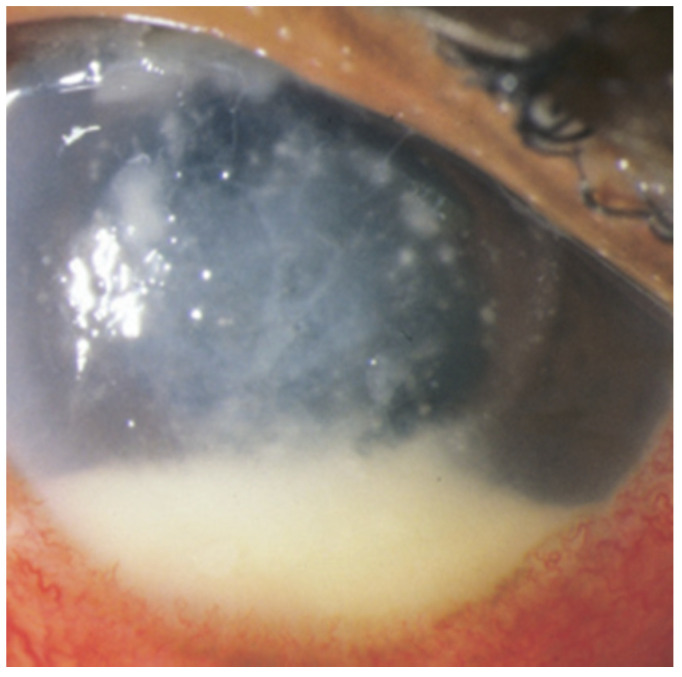
Photograph demonstrating *Fusarium oxysporum* endophthalmitis which developed 27 days after minor trauma with organic matter to the right eye. A hypopyon and anterior chamber fungal infiltrates are seen in the setting of a multifocal, feathery-edged corneal infiltrate. Adapted from Wykoff C.C., Flynn H.W., Jr., Miller D., Scott I.U., and Alfonso E.C. Exogenous fungal endophthalmitis: microbiology and clinical outcomes. *Ophthalmology*. 2008; 115(9): 1501–1507.e15072 [17]. Figure 1A, Copyright (2008) with permission from Elsevier. License 5138360747040 on 29 August 2021.

**Figure 2 jof-07-00996-f002:**
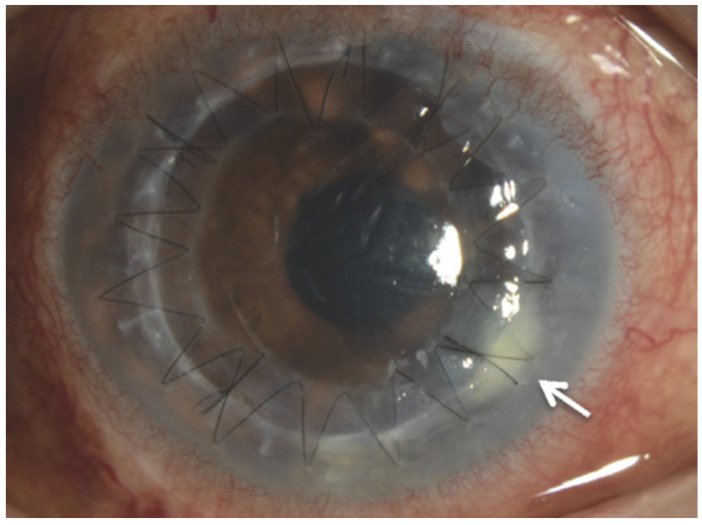
Slit-lamp examination showing that small white infiltrates were observed at the border between the host and donor corneal graft. Adapted from Kitazawa K., Wakimasu K., Yoneda K., Iliakis B., Sotozono C., and Kinoshita S. A case of fungal keratitis and endophthalmitis post penetrating keratoplasty resulting from fungal contamination of the donor cornea. *Am J Ophthalmol Case Rep*. 2016; 5: 103–106 [32]. Figure 2A, Copyright (2016) with permission from Elsevier. License 5135721109911 on 25 August 2021.

**Figure 3 jof-07-00996-f003:**
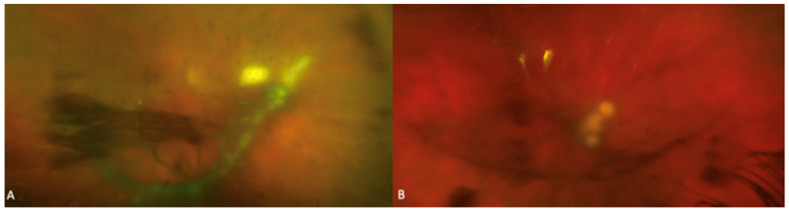
Fundus photographs of two patients with fungal endophthalmitis with (**A**) characteristic creamy-white and (**B**) well circumscribed “string of pearls” retinal and vitreous opacities.

**Figure 4 jof-07-00996-f004:**
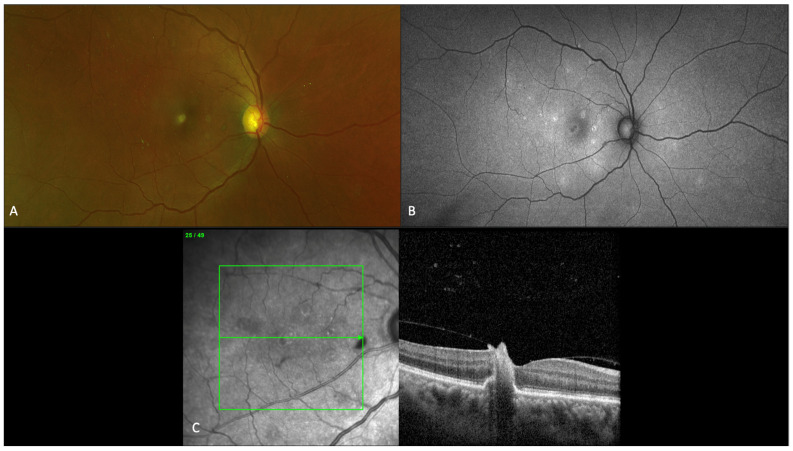
Imaging findings for a 56-year-old patient with fungal chorioretinitis, showing characteristic fundus, autofluorescence, and optical coherence tomography (OCT) macula findings. (**A**) Fundus photograph of right eye showing multiple small chorioretinal white lesions predominantly in the macula. (**B**) Fundus autofluorescence showing large area of hypoautofluoroescence in the fovea and multiple areas of scattered punctate hyperautofluoresence lesions in areas of early retinal pigment epithelium loss. (**C**) OCT macula findings over the foveal chorioretinal lesion depicting infiltration from choroid through retinal layers into the vitreous with focal areas of traction.

**Figure 5 jof-07-00996-f005:**
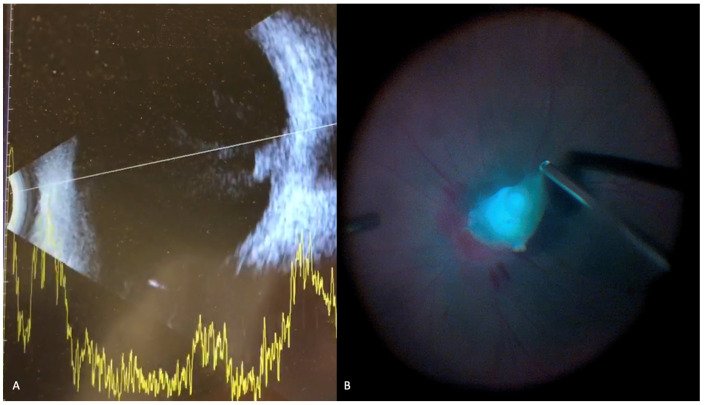
Ophthalmic B-scan showing (**A**) an endophytic lesion on optic nerve head and vitreous opacification (**B**) intra-operative view during surgical removal of large endophytic fungal ball that was previously identified on B-scan.

**Figure 6 jof-07-00996-f006:**
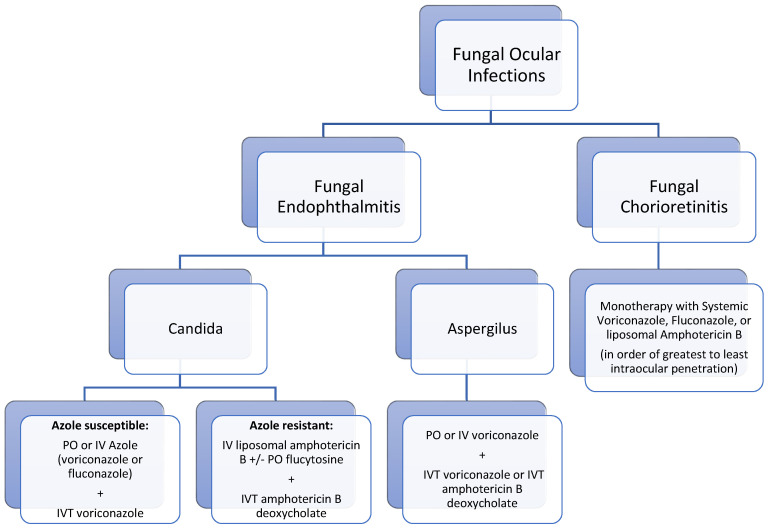
Algorithm for medical management of fungal endophthalmitis. Abbreviations: IVT, intravitreal; PO, oral; IV, intravenous. Systemic fluconazole: 800 mg (12 mg/kg) then 400–800 mg (6–12 mg/kg) daily. Systemic voriconazole: 400 mg (6 mg/kg) intravenous twice daily for 2 doses then 300 mg (4 mg/kg) intravenous or oral twice daily. Liposomal amphotericin B: 3–5 mg/kg intravenously daily, with or without oral flucytosine 25 mg/kg four times daily. Intravitreal amphotericin B: 5–10 µg/0.1 mL. Intravitreal voriconazole: 100 µg/0.1 mL.

**Table 1 jof-07-00996-t001:** Most common fungal etiologies in culture-positive post-traumatic endophthalmitis reported by Long et al. based on 912 reviewed cases [22].

Organism	Number (%)
Total Fungi	60 (100%)
*Aspergillus fumigatus*	11 (18.3%)
*Aspergillus nidulans*	7 (11.7%)
*Aspergillus niger*	6 (10%)
*Aspergillus flavus*	2 (3.3%)
*Fusarium solani*	5 (8.3%)
*Fusarium equiseti*	1 (1.7%)
*Fusarium moniliforme*	1 (1.7%)
*Bipolaris sorodiana*	4 (6.7%)
*Curvularia geniculate*	2 (3.3%)
*Conidia*	2 (3.3%)
*Penicilium*	1 (1.7%)

## Data Availability

Not applicable.

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
