# Peer review of "Fungal Endophthalmitis: A Comprehensive Review"

_jof, 2021, doi:10.3390/jof7110996_

Round 1

Reviewer 1 Report

This exhaustive review article clarifies the literature's deficiency on the lack of consensus on treatment strategy for fungal endophthalmitis. In addition, the authors could simplify the writing for the reader by using tables and algorithms.

The common fungi in traumatic endophthalmitis reported in the literature could be shown in a table, replacing the text at lines 90-99

Similarly, the authors could simplify the fungal endophthalmitis management by using treatment algorithms.

The authors should list the key differentiating features of fungal and bacterial endophthalmitis in a table.

The authors may suggest the best treatment combinations to optimize the therapy.

Minor corrections suggested -

Add citation to the line – line number 62-65

Citation mismatch at line 69

Rewrite lines 85 and 86

Punctuation error at line 543

Author Response

The common fungi in traumatic endophthalmitis reported in the literature could be shown in a table, replacing the text at lines 90-99

Fixed. See line 109.

Similarly, the authors could simplify the fungal endophthalmitis management by using treatment algorithms.

See line 562/Figure 6

The authors should list the key differentiating features of fungal and bacterial endophthalmitis in a table.

See line 427/Table 2

The authors may suggest the best treatment combinations to optimize the therapy.

See line 562/Figure 6

Minor corrections suggested -

Add citation to the line – line number 62-65

Fixed in line 65

Citation mismatch at line 69

Fixed in line 69

Rewrite lines 85 and 86

Rewritten in lines 86-89

Punctuation error at line 543

Fixed in line 572

Reviewer 2 Report

The present article is a broad yet concise review article of the pathogenesis, diagnosis, treatment and prognosis of fungal endophthalmitis. The authors did an excellent job at covering the pertinent materials to update the clinical and scientific community on an important and devastating disease in fungal endophthalmitis. The manuscript is well-written and well-organized. Other than a few minor comments below, I believe the review is well fit for publication.

Lines 36-37: such as “immunocompromise” not correct grammar. Should be something like “such as an immunocompromising condition”

Lines 84-85 Did the authors mean “the frequency of ENDOPHTHALMITIS in open globe injuries was 6.8%”?

Line 543: before contemporaneously there is an ( that is never closed.

Throughout the pandemic we have found at my local institute a dramatic rise in fungemia cases secondary to prolonged use intravascular catheters and intravenous steroid use. Have the authors found any data to suggest that fungal endophthalmitis cases have increased during the COVID 19 era?

Author Response

Lines 36-37: such as “immunocompromise” not correct grammar. Should be something like “such as an immunocompromising condition”

Fixed in line 37

Lines 84-85 Did the authors mean “the frequency of ENDOPHTHALMITIS in open globe injuries was 6.8%”?

Yes. Fixed in line 85

Line 543: before contemporaneously there is an ( that is never closed.

Fixed in line 572

Throughout the pandemic we have found at my local institute a dramatic rise in fungemia cases secondary to prolonged use intravascular catheters and intravenous steroid use. Have the authors found any data to suggest that fungal endophthalmitis cases have increased during the COVID 19 era?

Yes, we found some reports. Please see 239-248.

Round 2

Reviewer 1 Report

Reviewer's queries answered well